# Glance-and-Gaze Vision Transformer

**Qihang Yu[1], Yingda Xia[1], Yutong Bai[1], Yongyi Lu[1], Alan Yuille[1], Wei Shen[2]** *
[1] Department of Computer Science, Johns Hopkins University
[2] MoE Key Lab of Artificial Intelligence, AI Institute, Shanghai Jiao Tong University
`{yucornetto, philyingdaxia, ytongbai, yylu1989, alan.l.yuille}@gmail.com`
`wei.shen@sjtu.edu.cn`

## Abstract

Recently, there emerges a series of vision Transformers, which show superior performance with a more compact model size than conventional convolutional neural networks, thanks to the strong ability of Transformers to model long-range dependencies. However, the advantages of vision Transformers also come with a price: Self-attention, the core part of Transformer, has a quadratic complexity to the input sequence length. This leads to a dramatic increase of computation and memory cost with the increase of sequence length, thus introducing difficulties when applying Transformers to the vision tasks that require dense predictions based on high-resolution feature maps.

In this paper, we propose a new vision Transformer, named Glance-and-Gaze Transformer (GG-Transformer), to address the aforementioned issues. It is motivated by the Glance and Gaze behavior of human beings when recognizing objects in natural scenes, with the ability to efficiently model both long-range dependencies and local context. In GG-Transformer, the Glance and Gaze behavior is realized by two parallel branches: The Glance branch is achieved by performing self-attention on the adaptively-dilated partitions of the input, which leads to a linear complexity while still enjoying a global receptive field; The Gaze branch is implemented by a simple depth-wise convolutional layer, which compensates local image context to the features obtained by the Glance mechanism. We empirically demonstrate our method achieves consistently superior performance over previous state-of-the-art Transformers on various vision tasks and benchmarks.

## 1   Introduction

Convolution Neural Networks (CNNs) have been dominating the field of computer vision, which have been a de-facto standard and achieved tremendous success in various tasks, *e.g.*, image classification [16], object detection [15], semantic segmentation [5], *etc*. CNNs model images from a local-to-global perspective, starting with extracting local features such as edges and textures, and forming high-level semantic concepts gradually. Although CNNs prove to be successful for various vision tasks, they lack the ability to globally represent long-range dependencies. To compensate a global view to CNN, researchers explored different methods such as non-local operation [36], self-attention [33], Atrous Spatial Pyramid Pooling (ASPP) [5].

Recently, another type of networks with stacked Transformer blocks emerged. Unlike CNNs, Transformers naturally learn global features in a parameter-free manner, which makes them stronger alternatives and raises questions about the necessity of CNNs in vision systems. Since the advent of Vision Transformer (ViT) [12], which applied Transformers to vision tasks by projecting and tokenizing natural images into sequences, various improvements have been introduced rapidly, *e.g.*,

---

*Corresponding Author.

35th Conference on Neural Information Processing Systems (NeurIPS 2021).

better training and distillation strategies [32], tokenization [41], position encoding [7], local feature learning [14]. Moreover, besides Transformers' success on image classification, many efforts have been made to explore Transformers for various down-stream vision tasks [35, 24, 13, 3, 46].

Nevertheless, the advantages of Transformers come at a price. Since self-attention operates on the whole sequences, it incurs much more memory and computation costs than convolution, especially when it comes to natural images, whose lengths are usually much longer than word sequences, if treating each pixel as a token . Therefore, most existing works have to adopt a compromised strategy to embed a large image patch for each token, although treating smaller patches for tokens leads to a better performance (*e.g.*, ViT-32 compared to ViT-16 [12]). To address this dilemma, various strategies have been proposed. For instance, Pyramid Vision Transformer (PVT) [35] introduced a progressive shrinking pyramid to reduce the sequence length of the Transformer with the increase of network depth, and adopted spatial-reduction attention, where *key* and *value* in the attention module are down-sampled to a lower resolution. Swin-Transformer [24] also adopted the pyramid structure, and further proposed to divide input feature maps into different fix-sized local windows, so that self-attention is computed within each window, which reduces the computation cost and makes it scalable to large image scales with linear complexity.

Nonetheless, we notice that these strategies have some limitations: Spatial-reduction attention can reduce memory and computation costs to learn high-resolution feature maps, yet with a price of losing details which are expected from the high-resolution feature maps. Adopting self-attention within local windows is efficient with linear complexity, but it sacrifices the most significant advantage of Transformers in modeling long-range dependencies.

To address these limitations, we propose **Glance-and-Gaze Transformer (GG-Transformer)**, inspired by the Glance-and-Gaze human behavior when recognizing objects in natural scenes [11], which takes advantage of both the long-range dependency modeling ability of Transformers and locality of convolutions in a complementary manner. A GG-Transformer block consists of two parallel branches: A Glance branch performs self-attention within adaptively-dilated partitions of input images or feature maps, which preserves the global receptive field of the self-attention operation, meanwhile reduces its computation cost to a linear complexity as local window attention [24] does; A Gaze branch compensates locality to the features obtained by the Glance branch, which is implemented by a light-weight depth-wise convolutional layer. A merging operation finally re-arranges the points in each partition to their original locations, ensuring that the output of the GG-Transformer block has the same size as the input. We evaluate GG-Transformer on several vision tasks and benchmarks including image classification on ImageNet [10], object detection on COCO [23], and semantic segmentation on ADE20K [48], and show its efficiency and superior performance, compared to previous state-of-the-art Transformers.

## 2   Related Work

**CNN and self-attention.** Convolution has been the basic unit in deep neural networks for computer vision problems. Since standard CNN blocks were proposed in [22], researchers have been working on designing stronger and more efficient network architectures, *e.g.*, VGG [30], ResNet [16], MobileNet [29], and EfficientNet [31]. In addition to studying how to organize convolutional blocks into a network, several variants of the convolution layer have also been proposed, *e.g.*, group convolution [21], depth-wise convolution [6], and dilated convolution [40]. With the development of CNN architectures, researchers also seeked to improve contextual representation of CNNs. Representative works, such as ASPP [5] and PPM [45] enhance CNNs with multi-scale context, and NLNet [36] and CCNet [20] provided a non-local mechanism to CNNs. Moreover, instead of just using them as an add-on to CNNs, some works explored to use attention modules to replace convolutional blocks [18, 28, 34, 44].

**Vision Transformer.** Recently, ViT [12] was proposed to adapt the Transformer [33] for image recognition by tokenizing and flattening 2D images into sequence of tokens. Since then, many works have been done to improve Transformers, making them more suitable for vision tasks. These works can be roughly categorized into three types: (1) **Type I** made efforts to improve the ViT design itself. For example, DeiT [32] introduced a training scheme to get rid of large-scale pre-training and distillation method to further improve the performance. T2T-ViT [41] presented a token-to-token operation as alternatives to patch embedding, which keeps better local details. (2) **Type II** tried to

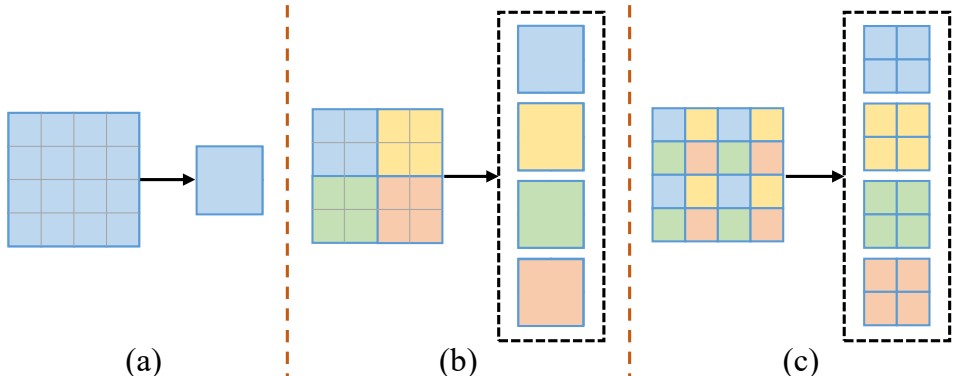

Figure 1: Toy examples illustrating different methods to reduce computation and memory cost of self-attention. (a) Spatial reduction [35, 13] spatially downsamples the feature map; (b) Local window [24] restricts self-attention inside local windows; (c) Glance attention (ours) applies self-attention to adaptively-dilated partitions.

introduce convolution back into the ViT design. *E.g.*, Chu *et al.* [7] proposed to use convolution for position encoding. Wu *et al.* [37] used convolution to replace the linear projection layers in Transformers. (3) **Type III** tried to replace CNNs by building hierarchical Transformers as a plug-in backbone in many downstream tasks. Wang *et al.* [35] proposed a pyramid vision Transformer, which gradually downsamples the feature map and extract multi-scale features as common CNN backbones do. However, applying self-attention on high-resolution features is not affordable in terms of both memory and computation cost, thus they used spatial-reduction attention, which downsamples *key* and *value* in self-attention as a trade-off between efficiency and accuracy. Later, Liu *et al.* [24] proposed a new hierarchical Transformer architecture, named Swin-Transformer. To handle the expensive computation burden incurred with self-attention, they divided feature maps into several non-overlapped windows, and limited the self-attention operation to be performed within each window. By doing so, Swin-Transformer is more efficient and also scalable to large resolution input. Besides, to compensate the missing global information, a shifted window strategy is proposed to exchange information between different windows.

Our method differs from aforementioned works in the following aspects: Type I, II methods usually utilize a large patch size and thus incompatible to work with high-resolution feature map. Type III methods proposed new attention mechanism to handle the extreme memory and computation burden with long sequences, but they sacrifices accuracy as a trade-off with efficiency. In contrast, GG-Transformer proposes a more efficient Transformer block with a novel Glance-and-Gaze mechanism, which not only enables it to handle long sequences and scalable to high-resolution feature maps, but also leads to a better performance than other efficient alternatives.

## 3 Method

The design of GG-Transformer draws inspiration from how human beings observe the world, which follows the *Glance and Gaze* mechanism. Specifically, humans will glance at the global view, and meanwhile gaze into local details to obtain a comprehensive understanding to the environment. We note that these behaviors surprisingly match the property of self-attention and convolution, which models long-range dependencies and local context, respectively. Inspired from this, we propose GG-Transformer, whose Transformer block consists of two parallel branches: A Glance branch, where self-attention is performed to adaptively-dilated partitions of the input, and a Gaze branch, where a depth-wise convolutional layer is adopted to capture the local patterns.

### 3.1 Revisit Vision Transformer

We start with revisiting the formulation of vision Transformer block, which consists of multi-head self-attention (MSA), layer normalization (LN), and multi-layer perceptron (MLP). A Transformer

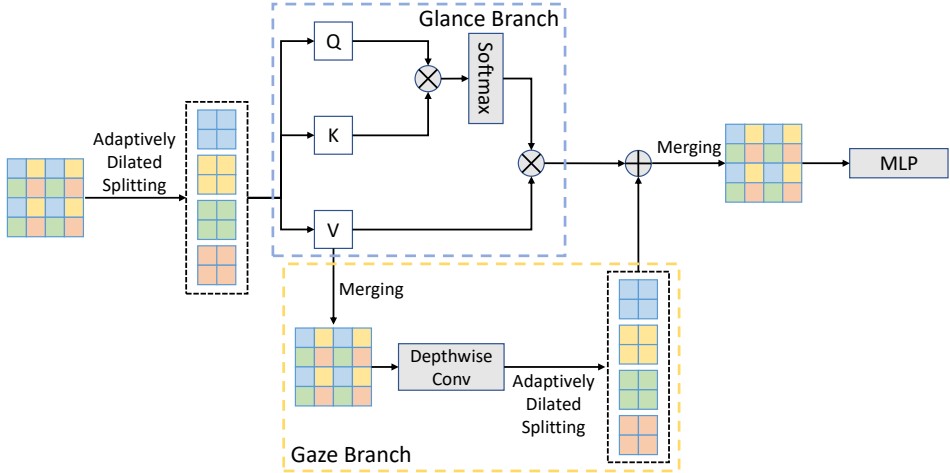

Figure 2: A visual illustration of GG Transformer block, where the Glance and Gaze branches parallely extract complementary information.

block processes input features as follows:

$$\mathbf{z}'_\ell = \mathrm{MSA}(\mathrm{LN}(\mathbf{z}_{\ell-1})) + \mathbf{z}_{\ell-1}, \tag{1}$$

$$\mathbf{z}_\ell = \mathrm{MLP}(\mathrm{LN}(\mathbf{z}'_\ell)) + \mathbf{z}'_\ell, \tag{2}$$

where $\mathbf{z}_\ell$ is the encoded image representation at the $\ell$-th block. MSA gives Transformers the advantages of modeling a global relationship in a parameter-free manner, which is formulated as:

$$\mathrm{MSA}(X) = \mathrm{Softmax}(\frac{QK^T}{\sqrt{C}})V, \tag{3}$$

where $Q, K, V \in \mathbf{R}^{N \times C}$ are the *query*, *key*, and *value* matrices which are linear mappings of input $X \in \mathbf{R}^{N \times C}$, $C$ is the channel dimension of the input, and $N$ is the length of input sequence. Note that for simplified derivation, we assume the number of heads is 1 in the multi-head self attention, which will not affect the following complexity analysis and can be easily generalize to more complex cases.

For vision tasks, $N$ is often related to the input height $H$ and width $W$. In practice, a 2D image is often first tokenized based on non-overlapping image patch grids, which maps a 2D input with size $H \times W$ into a sequence of token embeddings with length $N = \frac{H \times W}{P^2}$, where $(P, P)$ is the grid size. In MSA, the relationships between a token and all tokens are computed. Such designs, though effectively capturing long-range features, incur a computation complexity quadratic to $N$:

$$\Omega(\mathrm{MSA}) = 4NC^2 + 2N^2C. \tag{4}$$

For ViTs that only work on $16\times$ down-sampled feature maps (*i.e.*, $P = 16$), the computation cost is affordable, since in this scenario $N = 14 \times 14 = 196$ (for a typical ImageNet setting with input size $224 \times 224$). However, when it comes to a more general vision scenario with the need of dense prediction based on high-resolution feature maps (such as semantic segmentation), where the cost dramatically increases by thousands of times or even more. Naively applying MSA to such high-resolution feature maps can easily lead to the problem of out-of-memory (OOM), and extremely high computational cost. Although some efficient alternatives [35, 24] were brought up recently, accuracy is often sacrificed as a trade-off of efficiency. To address this issue, we propose a new vision Transformer that can be applied to longer sequence while keeping high accuracy, inspired by the Glance-and-Gaze human behavior when recognizing objects in natural scenes [11].

### 3.2 Glance: Efficient Global Modeling with Adaptively-dilated Splitting

To address the efficiency problem of Transformers, existing solutions often adapt Transformers to high-resolution feature maps by down-sampling the key and value during the attention process [35], or limit self-attention to be computed in a local region then exchange information through shifting these

local regions to mimic a global view [24]. But limitations exist in these methods. For down-sampling methods, although the output feature maps keep to be high-resolution, they lose some details during the down-sampling processes. Besides, they still has a quadratic complexity and thus may not scale up to a larger input size. For local-region based methods, though they successfully reduce the complexity to a linear level, they cannot directly model a long-range dependency but instead are stuck within local context, which counters the design intuition of Transformer and self-attention for long-range dependency modeling. Besides, two consecutive blocks need to work together to mimic a global receptive field, which may not achieve as good performance as MSA (see Table. 4b).

Thus, we propose Glance attention, which performs self-attention efficiently with a global receptive field. It shares same time complexity as [24], but directly models long-range dependencies, as shown in Fig. 1. Specifically, we first splits an input feature map to several dilated partitions, *i.e.*, the points in a partition are not from a local region but from the whole input feature map with a dilation rate adaptive to the feature map size and the token size. We name this operation *Adaptively-dilated Splitting*. For example, a partition contains $M \times M$ tokens and it is obtained with an adaptive dilation rate $= (\frac{h}{M}, \frac{w}{M})$, where $h$, $w$ is the height and width of current feature map respectively, and $hw = N$. Here we assume all divisions have no remainder for simplicity. These partitions are easily to be split from the input feature map or merged back. Specifically, we formulate this process as follows:

$$\mathbf{z}_{\ell-1} = [\mathbf{z}_{\ell-1}^{1,1}, \mathbf{z}_{\ell-1}^{1,2}, \ldots, \mathbf{z}_{\ell-1}^{h,w}], \tag{5}$$

where $\mathbf{z}_{\ell-1}^{i,j}$ is the feature token at position $(i, j)$ if reshaping the sequence of token embedding $\mathbf{z}_{\ell-1}$ back into a 2D feature map and use the 2D coordinates accordingly. To reduce the memory and computation burden, while keeping a global receptive field, $\mathbf{z}_{\ell-1}$ is split into several partitions:

$$\mathbf{z}_{\ell-1}^{\dagger} = \text{AdaptivelyDilatedSplitting}(\mathbf{z}_{\ell-1}) \tag{6}$$

$$= [\mathbf{z}_{\ell-1}^{\dagger,1,1}, \mathbf{z}_{\ell-1}^{\dagger,1,2}, \ldots, \mathbf{z}_{\ell-1}^{\dagger, \frac{h}{M}, \frac{w}{M}}], \tag{7}$$

$$\text{where } \mathbf{z}_{\ell-1}^{\dagger,i,j} = [\mathbf{z}_{\ell-1}^{i,j}, \mathbf{z}_{\ell-1}^{i,j+\frac{h}{M}}, \ldots, \mathbf{z}_{\ell-1}^{i+\frac{(M-1)h}{M}, j+\frac{(M-1)w}{M}}], \tag{8}$$

where $\mathbf{z}_{\ell-1}^{\dagger} \in \mathbf{R}^{N \times C}$, $\mathbf{z}_{\ell-1}^{\dagger,i,j} \in \mathbf{R}^{M^2 \times C}$. Afterwards, MSA is applied to each partition, which substantially reduces the computation and memory cost yet does not lose the global feature representation. And then all partitions are merged back into one feature map and go through the remaining modules:

$$\mathbf{z}_{\ell}^{\prime,i,j} = \text{MSA}(\text{LN}(\mathbf{z}_{\ell-1}^{\dagger,i,j})) + \mathbf{z}_{\ell-1}^{\dagger,i,j}, \tag{9}$$

$$\mathbf{z}_{\ell}^{\prime} = \text{Merging}(\mathbf{z}_{\ell}^{\prime,1,1}, \ldots, \mathbf{z}_{\ell}^{\prime, \frac{h}{M}, \frac{w}{M}}), \tag{10}$$

$$\mathbf{z}_{\ell} = \text{MLP}(\text{LN}(\mathbf{z}_{\ell}^{\prime})) + \mathbf{z}_{\ell}^{\prime}, \tag{11}$$

where $\text{Merging}$ is an inverse operation of $\text{AdaptivelyDilatedSplitting}$ which re-arranges points in each partition back in their original orders.

This new self-attention module (formulated in Eq. 6 to 10), namely Glance multi-head self attention module (G-MSA), enables a global feature learning with linear complexity:

$$\Omega(\text{G-MSA}) = 4hwC^2 + 2M^2hwC = 4NC^2 + 2M^2NC. \tag{12}$$

### 3.3 Gaze: Compensating Local Relationship with Depthwise Convolution

Although the Glance branch can effectively capture long-range representations, it misses the local connections across partitions, which can be crucial for vision tasks relying on local cues. To this end, we propose a Gaze branch to compensate the missing relationship and enhance the modeling power at a negligible cost.

Specifically, to compensate the local patterns missed in the Glance branch, we propose to apply an additional depthwise convolution on the *value* in G-MSA:

$$\text{Gaze}(X) = \text{DepthwiseConv2d}(\text{Merging}(V)), \tag{13}$$

which has a neglectable computational cost and thus the overall cost is still significantly reduced:

$$\Omega(\text{GG-MSA}) = 4NC^2 + 2M^2NC + k^2NC, \tag{14}$$

where $k$ is the Gaze branch kernel size, and $M$ is the partition size set in Glance branch, both $k$ and $M$ are constants that are much smaller than $N$. We note that in this way, long-range and short-range features are naturally and effectively learned. Besides, unlike [24], GG-MSA does not require two consecutive blocks (*e.g.*, W-MSA and SW-MSA) to be always used together, instead, it is a standalone module as the original MSA [12].

We propose two ways to determine the kernel size $k$ for better compensating the local features:

**Fixed Gazing.** A straightforward way is to adopt the same kernel size (*e.g.*, $3 \times 3$) for all Gaze branches, which can ensure same local feature learning regardless of the dilation rate.

**Adaptive Gazing.** Another way is implementing Gazing branch with adaptive kernels, where the kernel size should be the same as dilation rate $(h/M, w/M)$. In this way, GG-MSA still enjoys a complete view of the input.

By combining Glance and Gaze branches together, GG-MSA can achieve superior performance to other counterparts while remaining a low cost.

### 3.4 Network Instantiation

We build a hierarchical GG-Transformer with the proposed Glance-and-Gaze branches as shown in Fig. 2. For fair comparison, we follow the same settings as Swin-Transformer [24] in terms of network depth and width, with only difference in the attention methods used in Transformer blocks. Furthermore, we set $M$ to be same as the window size in [24], so that the model size and computation cost are also directly comparable. Note that GG-Transformer has not been specifically tuned by scaling depth and width for a better accuracy-cost trade-off.

We build GG-T and GG-S, which share the same model size and computation costs as Swin-T and Swin-S, respectively. For all GG-Transformers, we set the fixed patch size $M = 7$, expansion ratio of MLP $\alpha = 4$. All GG-Transformer consists of 4 hierarchical stages, which corresponds to feature maps with down-sampling ratio 4, 8, 16, 32, respectively. The first patch embedding layer projects input to a feature map with channel $C = 96$. When transitioning from one stage to the next one, we follow CNN design principles [16] to expand the channel by $2\times$ when the spatial size is down-sampled by $4\times$.

## 4 Experiments

In the following parts, we report results on ImageNet [10] classification, COCO [23] object detection, and ADE20K [48] semantic segmentation to compare GG-Transformer with those state-of-the-art CNNs and ViTs. Afterwards, we conduct ablation studies to verify the design of Glance and Gaze branches and also compare effectiveness of different alternative self-attention designs.

### 4.1 ImageNet Classification

We validate the performance of GG-Transformer on ImageNet-1K [10] classification task, which contains 1.28M training images and 50K validation images for 1000 classes. We report top-1 accuracy with a single $224 \times 224$ crop.

**Implementation Details.** To ensure a fair comparison, we follow the same training settings of [24]. Specifically, we use AdamW [25] optimizer for 300 epochs with cosine learning rate decay including 20 epochs for linear warm-up. The training batch size is 1024 with 8 GPUs. Initial learning rate starts at 0.001, and weight decay is 0.05. Augmentations and regularizations setting follows [32] including rand-augment [9], mixup [43], cutmix [42], random erasing [47], stochastic depth [19], but excluding repeated repeated augmentation [17] and EMA [26].

**Results.** A summary of results in Table 1, where we compare GG-Transformer with various CNNs and ViTs. It is shown that GG-Transformer achieve better accuracy-cost trade-off compared to other models. Moreover, GG-T, a light-weight model (28M/4.5G/82.0%), can achieve comparable performance to those even much large models such as DeiT-B (86M/17.5G/81.8%), T2T-ViT-24 (64M/14.1G/82.3%), and PVT-Large (61M/9.8G/81.7%). Furthermore, compared to Swin-Transformer, which we follows the architecture and ensures the same model size and computa-

Table 1: Comparison of different models on ImageNet-1K classification.

| method | image size | #param. | FLOPs | ImageNet top-1 acc. |
|---|---|---|---|---|
| RegNetY-4G [27] | $224^2$ | 21M | 4.0G | 80.0 |
| RegNetY-8G [27] | $224^2$ | 39M | 8.0G | 81.7 |
| RegNetY-16G [27] | $224^2$ | 84M | 16.0G | 82.9 |
| EffNet-B3 [31] | $300^2$ | 12M | 1.8G | 81.6 |
| EffNet-B4 [31] | $380^2$ | 19M | 4.2G | 82.9 |
| EffNet-B5 [31] | $456^2$ | 30M | 9.9G | 83.6 |
| DeiT-T [32] | $224^2$ | 5M | 1.3G | 72.2 |
| DeiT-S [32] | $224^2$ | 22M | 4.6G | 79.8 |
| DeiT-B [32] | $224^2$ | 86M | 17.5G | 81.8 |
| TNT-S [14] | $224^2$ | 24M | 5.2G | 81.3 |
| TNS-B [14] | $224^2$ | 66M | 14.1G | 82.8 |
| T2T-ViT-7 [41] | $224^2$ | 4M | 1.2G | 71.7 |
| T2T-ViT-14 [41] | $224^2$ | 22M | 5.2G | 81.5 |
| T2T-ViT-24 [41] | $224^2$ | 64M | 14.1G | 82.3 |
| PVT-Tiny [35] | $224^2$ | 13M | 1.9G | 75.1 |
| PVT-Small [35] | $224^2$ | 25M | 3.8G | 79.8 |
| PVT-Medium [35] | $224^2$ | 44M | 6.7G | 81.2 |
| PVT-Large [35] | $224^2$ | 61M | 9.8G | 81.7 |
| Swin-T [24] | $224^2$ | 28M | 4.5G | 81.2 |
| Swin-S [24] | $224^2$ | 50M | 8.7G | 83.2 |
| **GG-T** | $224^2$ | 28M | 4.5G | 82.0 |
| **GG-S** | $224^2$ | 50M | 8.7G | 83.4 |

tion costs to ensure a fair comparison, our model consistently brings an improvement to baseline, with a consistent improvement of 0.8% and 0.2% for T and S models respectively.

## 4.2 ADE20K Semantic Segmentation

ADE20K [48] is a challenging semantic segmentation dataset, containing 20K images for training and 2K images for validation. We follow common practices to use the training set for training and report mIoU results on the validation sets. We use UperNet [38] as the segmentation framework and replace the backbone with GG-Transformer.

**Implementation Details.** We follow [24] and use MMSegmentation [8] to implement all related experiments. We use AdamW [25] with a learning rate starting at $6 \times 10^{-5}$, weight decay of 0.01, batch size of 16, crop size of $512 \times 512$. The learning rate schedule contains a warmup of 1500 iterations and linear learning rate decay. The training is conducted with 8 GPUs and the training procedure lasts for 160K iterations in total. The augmentations follows the default setting of MMSegmentation, including random horizontal flipping, random re-scaling within ratio range [0.5, 2.0] and random photometric distortion. For testing, we follow [46] to utilize a sliding window manner with crop size 512 and stride 341. ImageNet-1K pretrained weights are used for initialization.

**Results.** We show results in Table 2, where results both w/ and w/o test-time augmentation are reported. Noticeably, GG-Transformer not only achieves better results to baselines, but also obtain a comparable single-scale testing performance to those with multi-scale testing results. Specifically, GG-T achieves 46.4% mIoU with single-scale testing, which surpasses ResNet50, PVT-Small, Swin-T's multi-scale testing results by 3.6%, 1.6%, 0.6%, respectively. Moreover, our tiny model even can be comparable to those much larger models (*e.g.*, 47.2% of GG-T compared to 47.6% of Swin-S).

## 4.3 COCO Object Detection

We further verify the performance of GG-Transformer when used as a plug-in backbone to object detection task on COCO dataset [23], which contains 118K, 5K, 20K images for training, validation and test respectively. We use Mask-RCNN [15] and Cascaded Mask R-CNN [1] as the detection frameworks, and compare GG-Transformer to various CNN and ViT backbones.

Table 2: Performance comparisons with different backbones on ADE20K validation dataset. FLOPs is tested on 1024×1024 resolution. All backbones are pretrained on ImageNet-1k.

| Backbone | UperNet | | | |
|---|---|---|---|---|
| | Prams (M) | FLOPs (G) | mIoU (%) | mIoU(ms+flip) (%) |
| ResNet50 [16] | 67 | 952 | 42.1 | 42.8 |
| PVT-Small [35] | 55 | 919 | 43.9 | 44.8 |
| Swin-T [24] | 60 | 941 | 44.5 | 45.8 |
| GG-T (ours) | 60 | 942 | 46.4 | 47.2 |
| ResNet101 [16] | 86 | 1029 | 43.8 | 44.9 |
| PVT-Medium [35] | 74 | 977 | 44.9 | 45.3 |
| Swin-S [24] | 81 | 1034 | 47.6 | 49.5 |
| GG-S (ours) | 81 | 1035 | 48.4 | 49.6 |

Table 3: Object detection and instance segmentation performance on the COCO `val2017` dataset using the Mask R-CNN framework. Params/FLOPs is evaluated with Mask R-CNN architecture on a 1280×800 image.

| Backbone | Params (M) | FLOPs (G) | Mask R-CNN | | | | | | Cascaded Mask R-CNN | | | | | |
|---|---|---|---|---|---|---|---|---|---|---|---|---|---|---|
| | | | $AP^b$ | $AP^b_{50}$ | $AP^b_{75}$ | $AP^m$ | $AP^m_{50}$ | $AP^m_{75}$ | $AP^b$ | $AP^b_{50}$ | $AP^b_{75}$ | $AP^m$ | $AP^m_{50}$ | $AP^m_{75}$ |
| ResNet50 [16] | 44 | 260 | 38.2 | 58.8 | 41.4 | 34.7 | 55.7 | 37.2 | 41.2 | 59.4 | 45.0 | 35.9 | 56.6 | 38.4 |
| PVT-Small [35] | 44 | 245 | 40.4 | 62.9 | 43.8 | 37.8 | 60.1 | 40.3 | - | - | - | - | - | - |
| Swin-T [24] | 48 | 264 | 43.7 | 66.6 | 47.7 | 39.8 | 63.3 | 42.7 | 48.1 | 67.1 | 52.2 | 41.7 | 64.4 | 45.0 |
| GG-T (ours) | 48 | 265 | 44.1 | 66.7 | 48.3 | 39.9 | 63.3 | 42.4 | 48.4 | 67.4 | 52.3 | 41.9 | 64.5 | 45.0 |
| ResNet101 [16] | 63 | 336 | 40.0 | 60.6 | 44.0 | 36.1 | 57.5 | 38.6 | 42.9 | 61.0 | 46.6 | 37.3 | 58.2 | 40.1 |
| ResNeXt101-32×4d [39] | 63 | 340 | 41.9 | 62.5 | 45.9 | 37.5 | 59.4 | 40.2 | 44.3 | 62.8 | 48.4 | 38.3 | 59.7 | 41.2 |
| PVT-Medium [35] | 64 | 302 | 42.0 | 64.4 | 45.6 | 39.0 | 61.6 | 42.1 | - | - | - | - | - | - |
| Swin-S [24] | 69 | 354 | 45.4 | 67.9 | 49.6 | 41.4 | 65.1 | 44.6 | 49.7 | 68.8 | 53.8 | 42.8 | 66.0 | 46.4 |
| GG-S (ours) | 69 | 355 | 45.7 | 68.3 | 49.9 | 41.3 | 65.3 | 44.0 | 49.9 | 69.0 | 54.0 | 43.1 | 66.2 | 46.4 |

**Implementation Details.** We follow the setting of [24] and use MMDetection [4] to conduct all the experiments. We adopt multi-scale training [2], AdamW optimizer [25] with initial learning rate of 0.0001, weight decay of 0.05, batch size of 16. The training is conducted with 8 GPUs and a 1× schedule. All models are initialized with ImageNet-1K pretrained weights.

**Results.** As shown in Table 3, GG-Transformer achieves superior performance to other backbones in the two widely-used detection frameworks. Specifically, GG-T achieves 44.1 box AP and 39.9 mask AP, which surpasses both CNNs and other ViTs with a similar model size and computation costs. Compared with the state-of-the-art Swin-Transformer, GG-Transformer achieves better performance while keeping the same model size and computation costs for both T and S models.

## 4.4 Ablation Studies

In this part, we conduct ablation studies regarding to the designs of GG-Transformer. Meanwhile, we also compare among different efficient alternatives to MSA. Besides, we verify GG-MSA on another ViT architecture [32] to compare its capacity to MSA directly. We conduct all these experiments based on Swin-T [24] with 100 epochs training and DeiT [32] architectures with 300 epochs training.

**Kernel Choice of Gaze Branch.** We study the choice of Gaze branch in terms of fixed or adaptive mechanism. The kernel sizes for each stage and results are summarized in Table 4a, where we observe that both mechanisms work well. Using a larger kernel leads to a non-significant improvement. In contrast, adaptive manner leads to a slightly better performance. Considering the adaptive manner provides a complete view as the original MSA has, we choose it in our final design.

**Glance/Gaze Branch.** We study the necessity of both Glance and Gaze branches. Meanwhile, a comparison between different ways to conduct self-attention is also studied. Results are in Table 4b.

Swin-T [24] serves as the baseline for all variants, which achieves 78.50% top-1 accuracy on ImageNet validation set. Firstly, we note that the local window attention and shifted window attention (W&SW-MSA) in [24] although can significantly reduce the computation complexity and makes

Table 4: Ablation studies regarding GG-Transformer design and comparison among different self-attention mechanisms.

(a) Choices of Gaze Kernels.

| Gaze Kernel | Top-1 |
|---|---|
| Fixed-(3,3,3,3) | 80.28% |
| Fixed-(5,5,5,5) | 80.31% |
| Adaptive-(9,5,3,3) | 80.38% |

(b) Comparison among different self-attentions. Gaze (Conv) uses kernels of Fixed-(3,3,3,3).

| | Top-1 |
|---|---|
| W& SW-MSA [24] | 78.50% |
| MSA | 79.79% |
| Glance Only | 77.21% |
| Gaze Only | 76.76% |
| Glance+Gaze (Attn) | 79.07% |
| Glance+Gaze (Conv) | 80.28% |

(c) Applying GG-MSA to DeiT backbone.

| | Top-1 |
|---|---|
| DeiT-T | 72.2% |
| GG-DeiT-T | 73.8% |
| DeiT-S | 79.9% |
| GG-DeiT-S | 80.5% |

Transformer easier to scale-up, it sacrifices the accuracy and the combination of W&SW-MSA to mimic a global view is not as good as the original MSA. We replace the W&SW-MSA with MSA for all blocks in stage 3 and 4 (*i.e.*, stages with down-sampling rate 16 and 32), which leads to a 1.29% performance improvement, indicating there exists a significant performance gap between MSA and its efficient alternative. Notably, when adopting the proposed Glance and Gaze mechanism instead, which shares a same complexity of W& SW-MSA, can achieves much better performance, where the Glance+Gaze (Attn) improves the performance by 0.57%, and Glance+Gaze (Conv) (*i.e.*, GG-T) by 1.78%, which is even higher than MSA by 0.49%.

Besides using depthwise convolution, another natural choice is to also adopt self-attention for implementing the Gaze branch. Therefore, we conduct experiments by using local window attention [24] as the Gaze branch. Note that, unlike depthwise convolution, a self-attention variant of the Gaze branch cannot be integrated with the Glance branch into the same Transformer block while keeping the overall model size and computation cost at the same level. To ensure a fair comparison, we use two consecutive Transformer blocks where one is Glance attention and another is Gaze attention. Using either convolution or self-attention to implement the Gaze branch can both improve the performance compared to [24], illustrating the effectiveness of the Glance and Gaze designs. However, using self-attention is inferior to depth-wise convolution with a degrade of 1.21%, which may indicate that convolution is still a better choice when it comes to learning local relationships. Besides, using depth-wise convolution as Gaze branch can also naturally be integrated into the Transformer block with Glance attention, thus makes it more flexible in terms of network designs.

We also note that Glance or Gaze branch alone is far from enough, while only a combination of both can lead to a performance gain, which matches the behavior that we human beings can not rely on Glance or Gaze alone. For instance, using Glance alone can only lead to an inferior performance with accuracy of 77.21%, and Gaze alone 76.76%, which is significantly lower than baseline with a degrade of 1.29% and 1.74%, respectively. Nevertheless, we note that this is because Glance and Gaze branches miss important local or global cues which can be compensated by each other. As a result, a combination of both Glance and Gaze gives a high accuracy of 80.28%, which improves the Glance alone and Gaze alone by 3.07% and 3.52% respectively.

**Apply to other backbone.** We verify the effectiveness of GG-Transformer on another popular ViT architecture DeiT [32], as shown in Table 4c. We replace MSA with GG-MSA for two DeiT variants [32], DeiT-T and DeiT-S. We show that, although GG-MSA is an efficient alternative to MSA, it can also lead to a performance gain. Compared to DeiT-T and DeiT-S, GG-DeiT-T and GG-DeiT-S bring the performance up by 1.6% and 0.6% respectively, illustrating that it is not only efficient but also effectively even compared to a fully self-attention.

**Network runtime.** We follow [24] to report and compare work runtime measured by FPS: GG-T achieves 782.34 FPS compared to 737.86 of Swin-T, and GG-S achieves 441.31 FPS compared to 423.51 of Swin-S. The evaluation is done with a single Nvidia tesla v100-sxm2-16gb GPU.

## 5 Limitation

Although GG-Transformer provides a powerful and efficient solution to make Transformers scalable to large inputs, some limitations still exist and worth further exploring.

Firstly, over-fitting is a common problem [12] in Vision Transformers and can be alleviated by large-scale pretraining [12] or strong augmentations and regularization [32]. This problem is more serious for stronger models (GG-Transformer) and in the tasks with relatively small dataset (e.g. semantic segmentation). Secondly, Transformers suffer from performance degradation in modeling longer-range dependencies, when there exists large discrepancy in the training-testing image size. The limitations can come from position encoding, which has fixed size and need to be interpolated to different input sizes, or self-attention itself, which may not adapt well when significant changes happen in input size. Lastly, there is a long-lasting debate on the impacts of AI on human world. As a method improving the fundamental ability of deep learning, our work also advances the development of AI, which means there could be both beneficial and harmful influences depending on the users.

## 6 Conclusion

In this paper, we present GG-Transformer, which offers an efficient and effective solution to adapting Transformers for vision tasks. GG-Transformer, inspired by how human beings learn from the world, is equipped with parallel and complementary Glance branch and Gaze branch, which offer long-range relationship and short-range modeling, respectively. The two branches can specialize in their tasks and collaborate with each other, which leads to a much more efficient ViT design for vision tasks. Experiments on various architectures and benchmarks validate the advantages of GG-Transformer.

**Acknowledgment**: This work was supported by ONR N00014-20-1-2206, ONR N00014-18-1-2119, NSFC 62176159, Natural Science Foundation of Shanghai 21ZR1432200, SJTU Explore X Foundation, and Shanghai Municipal Science and Technology Major Project 2021SHZDZX0102.

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
