# OpenReview forum: "Glance-and-Gaze Vision Transformer"
_NeurIPS.cc/2021/Conference — NeurIPS 2021 Poster_

### Official Review · Reviewer_E8bD · 2021-07-14

**Rating:** 6
**Confidence:** 5

**Summary:**

This paper presents a glance and gaze module to replace standard self-attention with higher efficiency in terms of memory. The paper validated their methods in image classification, object detection, and semantic segmentation.

**Limitations And Societal Impact:**

Yes

**Main Review:**


1. This paper achieved a better trade-off between FLOPs and accuracy than vanilla ViT.

2. Runtime: Even though the proposed method has the same FLOPs as Swin; however, it seems that it needs more folding/unfolding operations which could slow down the network in runtime.


Questions

1. See above point 2. What is the runtime speed compared to Swin?

2. Why do the authors use stronger data augmentation in object detection? in PVT and Swin, for 1x schedule, they do not use multi-scale training. What is the performance if disabling multi-scale training?

3. Why the ablation study does not use the same settings, do those observations still hold if using the same settings?

4. Why the proposed model (Gaze + Glance) is 80.28% in Table 4b while 80.38% in Table 4a?

5. In Table 4c, what is the performance of DeiT without repeated augmentation?

6. Does the proposed method scale better? E.g. from Swin-S to Swin-B, the performance does not improve too much. How about the proposed method, GG-B?

7. It seems that the proposed method achieved better improvement for the smaller model only (GG-T) while the gain over Swin-S is limited. I wonder what is the explanation.



**Time Spent Reviewing:**

8

---

> ### Author Response · Authors · 2021-08-10
> **Authors' Response to Reviewer xdt9**
>
> We thank the reviewer for the detailed comments and valuable questions. Below we provide responses to each concern.
>
> **Q1: More folding/unfolding operations which could slow down the network in runtime.**
>
> A1: We note that in Swin-Transformer, to implement W/SW-MSA, reshape, permute, and shift operations are also needed, which has a similar cost to fold/unfold operations. We also note that, by properly optimizing the codes, we can reduce the number of calls of these operations as much as possible to obtain an even higher FPS, e.g., by postponing the Adaptively Dilated Splitting after QKV generation and move the final Merging to before MLP (in Fig. 2), we can save the unfolding/folding for Gaze branch but still remain mathematically the same. As a result, GG can even surpass the FPS of Swin. We report the FPS comparison as follows:
>
> |      |  Tiny  | Small  |
> |------|:------:|--------|
> | Swin | 737.86 | 423.51 |
> | GG   | 782.34 | 441.31 |
>
> The testing uses the official evaluation code of [24] (i.e., using ImageNet images, batch_size = 64, averaged on 30 runs) to measure the throughput.
>
> Environment: Nvidia tesla v100-sxm2-16gb, Intel(R) Xeon(R) CPU E5-2698 v4 @ 2.20GHz, PyTorch 1.7.1, CUDA 10.2. Please note that the number of Swin can be slightly different from reported in [24] due to enviroment/hardware differences.
>
> **Q2: Why use stronger data augmentation in object detection.**
>
> A2: The reviewer might misunderstand the setting for object detection. We strictly follow the same data augmentation setting used in the official code of Swin-Transformer[r1], i.e., Swin also adopts multi-scale training for 1x schedule. Without multi-scale training, GG-T performance achieves box mAP = 43.5 and mask mAP = 39.8, which is significantly higher than PVT-Small which is also trained without multi-scale training with box mAP = 40.4 and mask mAP = 37.8.
>
> **Q3: Why the ablation study does not use the same settings, do those observations still hold if using the same settings?**
>
> A3: To reduce computation budgets, we conduct ablation studies by training models with fewer iterations, i.e.100 epochs, which is a commonly-used setting for the ResNet family training and is still a reasonable setting to ablate different designs of GG-Transformer.
>
> **Q4: Why the proposed model (Gaze + Glance) is 80.28% in Table 4b while 80.38% in Table 4a?**
>
> A4: The results in Table 4b are based on the model with Gaze kernel Fixed-(3,3,3,3) instead of Adaptive-(9,5,3,3), thus the number is 80.28%, **which is correct**. We will update the Table caption to make it more clear.
>
> **Q5: In Table 4c, what is the performance of DeiT without repeated augmentation?**
>
> A5: As shown in Table 8 of [32], without repeated augmentation, the DeiT-B performance drops dramatically from 81.8 to 76.5. In our experiments, we always adopt the strongest setting, which includes repeated augmentation, for both DeiT and GG-DeiT, to ensure a fair comparison and the improvement is significant even on stronger baselines.
>
> **Q6: Scale-up to larger model, e.g., GG-B?**
>
> A6: Thanks for the good question. As the model scales up, the baseline gets stronger and it becomes more difficult to achieve a large improvement. We also note that from Swin-S to Swin-B, no significant improvement is observed, which we believe stronger augmentation/regularization should be introduced for Swin-B/GG-B. Due to computational resource limitations, we did not conduct experiments with GG-B but will do in the future.
>
> **Q7: Gain over tiny model v.s. small model.**
>
> A7: When the baseline gets stronger, it is harder to achieve a very large improvement. We also note that similar trends are observed in some other concurrent works [r2, r3] trying to improve Swin.
>
> [r1] https://github.com/SwinTransformer/Swin-Transformer-Object-Detection
>
> [r2] Xiangxiang Chu, Zhi Tian,  Yuqing Wang, Bo Zhang, Haibing Ren, Xiaolin Wei, Huaxia Xia, Chunhua Shen. Twins: Revisiting the Design of Spatial Attention in Vision Transformers. arXiv preprint arXiv:2104.13840, 2021.
>
> [r3] Jiemin Fang, Lingxi Xie, Xinggang Wang, Xiaopeng Zhang, Wenyu Liu, Qi Tian. MSG-Transformer: Exchanging Local Spatial Information by Manipulating Messenger Tokens. arXiv preprint arXiv:2105.15168, 2021.

---

> > ### Comment · Reviewer_E8bD · 2021-08-26
> > **Thanks for the rebuttal**
> >
> > Thanks for your feedback. The authors addressed most of my concerns.
> > For Q3, the number of epochs for ablation study, as you trained the model with 300 epochs for the final results, without training under the same setting, it is hard to see what is the exact benefit of the proposed module. E.g. maybe the proposed method can converge faster so it is benefited by training fewer epochs.
> > On the other hand, for Q5, this paper is talking about DeiT-Ti, not DeiT-B.
> > Even though I still have concerns, I upgraded my rate to 6.

---

### Official Review · Reviewer_xdt9 · 2021-07-16

**Rating:** 7
**Confidence:** 3

**Summary:**

The paper introduces a novel attention mechanism for the transformer architecture. Inspired by human gaze, authors propose to split the self-attention operation which is typically done across all patches in two parallel processes: one which reasons locally and another one that reasons globally. Authors evaluate their method in standard datasets for segmentation, classification and detection and ablate the main design decisions.

**Limitations And Societal Impact:**

Authors discuss limitations and societal impact of their work.

**Main Review:**

Summary: The paper introduces one single idea, describes it very well and show its effectiveness through evaluation in multiple tasks on standard benchmarks. For those reasons, I believe the paper should be accepted.

Strengths:
- I really like how the block splits the standard self-attention operation into local and global attention in a simple but effective way. I believe that as transformers are becoming a more relevant architecture over time, the community will need to move towards more efficient architectures to process images in transformers.

- Authors evaluate their model in three tasks: detection, segmentation and classification. I believe this makes the paper very strong, as it shows that the proposed block is effective not only for the standard classification task (which is no affected by local changes in the image) but also works well for detection and segmentation.

-   The results of the evaluation in the three task show that the performance of the proposed method improves previous proposal. Furthermore, authors report an extensive list of baselines which helps the reader put the result in perspective.

- Authors ablate the most relevant design decisions of the paper. Specially, I like that the block also improves performance when used in other backbones Table 4.c.

- The paper is very well written and explains very well the approach proposed.

Weaknesses:

- I am missing some qualitative evaluation on what each block is doing on the operation. I think this would be very interesting for the reader to gain a bit of understanding of the model. This could be archived by providing some qualitative results of the attention, or even examples where the proposed model improves the baselines.

- In L58, the sentence in italic is a bit confusing and it appears to be the name of a paper (which is not). I would recommend the authors edit that part of the paper to make this clearer.

- Have the authors considered processing the gaze part without a convolution and with a transformer-like approach? I would be curious to see if there a solution without involving convolution and how does that perform to the proposed method.

- I think more discussion about the computational reduction would be also interesting and make the paper stronger. For instance, how would this approach scales to large-resolution images compared to traditional transformer approaches (ViT).



**Time Spent Reviewing:**

3.5

---

> ### Author Response · Authors · 2021-08-10
> **Authors' Response to Reviewer xdt9**
>
> We thank the reviewer for the valuable comments and suggestions. And we address the concerns as follows.
>
> **Q1: Qualitative evaluation of what each block is doing on the operation.**
>
> A1: Thanks for the valuable suggestion! We totally agree that visualization of attention maps may help readers to better understand our work. Unfortunately, we are unable to give the illustration in the rebuttal as the image is not allowed. Thanks really for the suggestion.
>
> **Q2: Sentence in L58.**
>
> A2: Thanks for the suggestion! We will remove the italic font in a revision for better readability according to the suggestion.
>
> **Q3: Transformer-like approach instead of convolution as gaze branch.**
>
> A3: Thanks for the good question! We have done a related ablation study in Supplementary Table 2, where we have tried to use a local self-attention to implement the Gaze branch. It shows inferior performance compared to using convolution for the Gaze branch. The reason might be when the regions to be dealt with are relatively small and local details are more important, the advantage of self-attention in modeling longer-range context becomes weaker and thus convolution performs better.
>
> **Q4: Scale to larger-resolution and comparison to ViT.**
>
> A4: We have provided a theoretical computation complexity to compare the scale-up ability between GG-Transformer and ViT in Eq. (4, 12), where GG-MSA shows a linear complexity to input size compared to quadratic complexity in MSA of ViT. We acknowledge that comparison on high-resolution input could be better, yet it is practically difficult to train ViT with a much higher resolution due to the memory constraints.

---

> ### Comment · Reviewer_xdt9 · 2021-09-26
> **Updating review for visibility from the authors**
>
> Hi,
>
> After reading the rebuttal and the others reviews, I believe the paper should be accepted. I think the authors have addressed my concerns in their response, and therefore I think the paper has sufficient merits to be accepted. Furthermore, I have read the other reviews, and although I share some of the concerns, I think the author's response is overall convincing.
>
> Best wishes,

---

### Official Review · Reviewer_y1o2 · 2021-07-20

**Rating:** 3
**Confidence:** 3

**Summary:**

This paper proposes a Glance and Gaze vision transformer to overcome the complexity problem common in vision transformers. The proposed transformer has two parallel branches - a glance branch that accounts for long-term dependencies and a gaze branch to account for local context. The authors compare their work on ImageNet-1k, COCO and ADE20K data sets for image classification, object recognition and semantic segmentation. Reference [24] (Swin Transformer) was the prior art on which the experimental set-up was taken from. Comparisons with a number of other vision transformer settings were made.

**Ethical Concerns:**

There are no major ethical issues assuming there is a good explanation for the issue raised in comments 2 and 3.

**Limitations And Societal Impact:**

To some extent.

**Main Review:**

The idea of Glance and Gaze has an intuitive appeal and this is appreciated by the reviewer. However, there are a number of issues with this paper:

1) The paper has numerous grammatical errors and typos which made it hard to follow.
2) The main issue with this paper is that it is unable to beat the accuracy of [24] which they have taken as baseline. For example, consider the last four rows of Table 1 which compares Swin with the proposed GG. The complexity is identical and accuracy similar. Also, results with ImageNet-22k are not shown.
3) The Swin-T and Swin-S numbers listed in Table 2 for ADE20K task are lower than the ones reported in [24]. Why is this? The original paper [24] reports mIoU of 46.1% (Swin-T) and 49.3% (Swin-S) which are higher than those obtained by GG which further buttresses the comment in 2). FPS numbers are not reported. Not sure what is going on.
4) Table 3 for object detection has the same issue as in Table 2 (see Comment 3). The numbers reported by [24] are higher that those cited and compared with. Also, since [24] is the main prior work for establishing a comparison framework, there needs to be comparison with other object detection frameworks such as ATSS, Rev Points V2...

It is important that the authors clarify the issue of numbers from prior art they are quoting in the paper (see comments 3 and 4) and expand the set of comparisons they are making to match up with reference [24]. The scope of the paper needs to cover at least the same data sets and metrics as in [24].


**Time Spent Reviewing:**

1.5

---

> ### Author Response · Authors · 2021-08-10
> **Authors' Response to Reviewer y1o2**
>
> We thank the reviewer for the detailed comments. Below we provide responses to the concerns, where we aim to address major misunderstandings.
>
> **Q1: Numerous grammatical errors and typos.**
>
> A1: Since the other reviewers (G6eL, xdt9) both agree the paper is well-written, we think the grammatical errors and typos should not be too many to affect the readability. Of course, we will proofread and polish our paper carefully.
>
> **Q2: Complexity and performance compared with Swin-Transformer, ImageNet-22k?**
>
> A2: We emphasize that to ensure a fair comparison to our major baseline, Swin-Transformer, we adopt the exact same architecture and use their official codes for all experiments. Therefore, we designedly align the parameters/computational costs of GG-Transformer and those of Swin-Transformer on purpose to fairly compare GG-MSA with W/SW-MSA and to show that GG-MSA has stronger representation ability with the same cost, e.g., in Table 4(b) we use the same architecture but adopt different attention block with the same cost, and GG-MSA shows better performance to other counterparts. We believe that the architecture of GG-Transformer (e.g., partition size, network width, and depth) can be further tuned to obtain an even better accuracy-cost trade-off.
>
> Besides, the improvement (+0.8% on tiny and +0.2% on small) is significant especially considering the Swin is a very strong baseline and we exactly follow Swin's settings to ensure a fair comparison but did not finetune them for our model.
>
> For ImageNet-22k, we did not conduct the related experiments because the official code-base of Swin does not support training on ImageNet-22k, and also due to the limitation on computational resources, which can take up to half a month with an 8 GPU machine.
>
> **Q3: Numbers are wrong in Table 2 and Table 3?**
>
> A3: The reviewer might misunderstand the settings to obtain these results. All numbers are **correct** and are obtained based on Swin's official GitHub repo[r1, r2].
>
> The results of 46.1% (Swin-T) and 49.3% (Swin-S) reported in [24]  are obtained with test-time augmentation, which corresponds to the results of "mIoU(ms+flip)" in Table 2 in our paper, where we have 45.8% (Swin-T) and 49.5% (Swin-S). These results are directly quoted from Swin's official GitHub repo[r1] (the Table of "ADE20K" in "Results and Models"), which are almost the same as the results reported in [24].
>
> Similarly, in Table 3 in our paper, the results are obtained by training with the 1x schedule instead of the 3x schedule, as mentioned in L262-263. We report Swin's results based on the official repo[r2]. For Swin-T, since the official repo provides the results trained with the 1x schedule, we directly quote them: 43.7 box mAP and 39.8 mask mAP (the Table of "Mask R-CNN" in "Results and Models".). For Swin-S, since the official repo does not provide the results trained with the 1x schedule, we ran the official code by ourselves to produce the results.
>
> **To sum up, we have tried our best to ensure the fairness and correctness of all numbers. We quoted results from the official GitHub repo and use the same code-base of Swin-Transformer for all main experiments of GG-Transformer.**
>
> **Q4: FPS, more object detection frameworks.**
>
> A4: Thanks for pointing it out. We test Swin-T/S and GG-T/S using the official evaluation code of [24] (i.e., using ImageNet images, batch_size = 64, averaged on 30 runs) to measure the throughput:
>
> |      |  Tiny  | Small  |
> |------|:------:|--------|
> | Swin | 737.86 | 423.51 |
> | GG   | 782.34 | 441.31 |
>
> Environment: Nvidia tesla v100-sxm2-16gb, Intel(R) Xeon(R) CPU E5-2698 v4 @ 2.20GHz, PyTorch 1.7.1, CUDA 10.2. Please note that the number of Swin can be slightly different from reported in [24] due to enviroment/hardware differences.
>
> We did not apply GG-Transformer to all the detection frameworks due to computational resource limitations, and we will try out best to add them in the future.
>
>
>
> [r1] https://github.com/SwinTransformer/Swin-Transformer-Semantic-Segmentation
>
> [r2] https://github.com/SwinTransformer/Swin-Transformer-Object-Detection

---

### Official Review · Reviewer_G6eL · 2021-07-21

**Rating:** 7
**Confidence:** 4

**Summary:**

The paper presented a new Transformer-based model for visual recognition. The key idea is to use a checkerboard-like pattern (dilated splitting) to divide the 2D feature plan into multiple smaller partitions, where the standard self-attention can be efficiently computed on each of these partitions (glance branch). A local depthwise convolution was also added to link the partitions (gaze branch). To evaluate the proposed model, the authors presented extensive results across several visual recognition tasks, including image classification, semantic segmentation, and object detection. The results are promising.


**Limitations And Societal Impact:**

Yes

**Main Review:**

**Originality**

The key idea of the glance branch (dilated splitting) is well suited for 2D images and quite interesting. The combination of the glance / gaze branch is convincing.

My main concern is the missing link to previous works on efficient Transformer models in the ML / NLP community. Specifically, there has been quite some recent effort on developing linear time self-attention / Transformer blocks, e.g.,  Linformer [1], Longformer [2], Performer [3], Reformer [4], Nystromformer [5], Informer [6] and Big Bird [7]. As one of the main arguments of this paper is an efficient Transformer model, these previous works seem quite relevant and should be discussed and/or compared. My understanding is that the proposed method is tailored for 2D visual data, while previous approaches are more general and can also be applied to 2D data. Further, some of the ideas proposed in the paper can be found in these previous works.  For example, using sparse attention from different local pattern was discussed in [7] while using depthwise convolution in parallel to self-attention was presented in [5]. It thus remains unclear to me how the proposed method compares to these previous approaches.

[1] Wang, S., Li, B. Z., Khabsa, M., Fang, H., & Ma, H. (2020). Linformer: Self-attention with linear complexity. arXiv preprint arXiv:2006.04768.

[2] Beltagy, I., Peters, M. E., & Cohan, A. (2020). Longformer: The long-document transformer. arXiv preprint arXiv:2004.05150.

[3] Choromanski, K., Likhosherstov, V., Dohan, D., Song, X., Gane, A., Sarlos, T., ... & Weller, A. (2020). Rethinking attention with performers. In ICLR.

[4] Kitaev, N., Kaiser, Ł., & Levskaya, A. (2020). Reformer: The efficient transformer. In ICLR.

[5] Xiong, Y., Zeng, Z., Chakraborty, R., Tan, M., Fung, G., Li, Y., & Singh, V. (2021). Nystr\" omformer: A Nystr\" om-Based Algorithm for Approximating Self-Attention. In AAAI.

[6] Zhou, H., Zhang, S., Peng, J., Zhang, S., Li, J., Xiong, H., & Zhang, W. (2021, May). Informer: Beyond efficient transformer for long sequence time-series forecasting. In AAAI.

[7] Zaheer, M., Guruganesh, G., Dubey, K. A., Ainslie, J., Alberti, C., Ontanon, S., ... & Ahmed, A. (2020, July). Big Bird: Transformers for Longer Sequences. In NeurIPS.


**Quality**

The paper is technically sound. Experiments are thorough. And the results are solid.

A main motivation of the model is the visual attention in the human visual system. Yet I don’t see a clear link between self-attention (computation of self similarity between features) and human visual attention (selection of relevant sensory information). I do think that the idea of dilated splitting stems from dilated kernels in 2D convolution. And that seems to be a better perspective.

**Clarity**

The paper is reasonably well written. The key idea is nicely illustrated.

**Significance**

The paper addressed the design of vision Transformers, a trendy topic in the field. The proposed model has the potential to serve as a common backbone for many vision tasks. The downside is that the proposed model seems to only have minor performance gains and similar computational costs when compared to Swin Transformer in [24], though I recognize that [24] is an arXiv paper and not yet published.

**Post-rebuttal Update**

Thanks for the authors' response. I've read other reviews and responses. A main concern shared among reviewers is the comparison to Swin-Transformer [24]. The clarification of details and the additional results provided in the rebuttal have addressed my concern. Further, given that [24] appeared on arXiv around two months before the NeurIPS deadline, I don't think this paper should get penalized for not beating the results in [24]. That being said, I encourage the authors to clarify the experiment settings in the paper, and to improve the presentation quality. Overall, I think this is a paper with an interesting idea and good results and I am happy to increase my rating.

**Time Spent Reviewing:**

3

---

> ### Author Response · Authors · 2021-08-10
> **Authors' Response to Reviewer G6eL**
>
> We thank the reviewer for the detailed comments. Below we provide responses to each concern.
>
> **Q1: Comparison to efficient Transformers in ML/NLP community.**
>
> A1: Thanks for introducing these NLP works to us! We will cite them in the related work and discuss the relationship. Although it is commonly accepted that many vision Transformers are inspired from NLP Transformers, it is non-trivial to extend the ideas of NLP Transformers to the vision domain, since they may miss 2D local patterns. E.g., in Table 6 of [24], a trial of directly replacing Transformers with Performers significantly degrades the performance.
>
> **Q2: Dilated splitting and dilated kernels in 2D convolution.**
>
> A2: Thanks for the advice. Our idea originates from human behavior which observes the world in a coarse-to-fine manner.  In terms of implementation, we use sparse global attention to mimic the glance behavior, which captures the global view at the cost of losing details, and the convolution as a gaze branch to compensate for the missing details. We thank you for proposing the connection between dilated splitting and dilated kernels and will surely discuss them in a revision.
>
> **Q3: Minor performance gains and similar computational costs when compared to Swin-Transformer.**
>
> A3: We emphasize that to ensure a fair comparison to our major baseline, Swin-Transformer, we adopt exactly the same architecture and use their official codes for all experiments. Therefore, we designedly align the parameters/computational costs of GG-Transformer and those of Swin-Transformer on purpose to fairly compare GG-MSA with W/SW-MSA and to show that GG-MSA has stronger representation ability with the same cost, e.g., in Table 4(b) we use the same architecture but adopt different attention block with the same cost, and GG-MSA shows better performance to other counterparts. We also believe that the architecture of GG-Transformer (e.g., partition size, network width, and depth) can be further tuned to obtain an even better accuracy-cost trade-off.

---

### Decision · Program_Chairs · 2021-09-27

**Decision:**

Accept (Poster)

**Comment:**

The paper introduces an efficient transformer architecture with a “glance” branch to model long-range dependencies and a “gaze” branch to account for local context. Three reviewers recommend acceptance, highlighting that the idea is novel, the paper is well-written, and the results are solid. One reviewer, despite appreciating the intuitive appeal of the glance and gaze idea, recommends rejection primarily because of the discrepancy between the Swin baseline numbers being compared with and the actual numbers in the Swin paper referenced in the manuscript. However, the AC agrees with the other three reviewers and the authors based on their response that the comparison with Swin transformers is fair, as the same setting was used to compare both models. The paper is technically sound, has an interesting idea, and the results are promising. The authors should carefully proofread the paper and add the discussion in the rebuttal to the final version.